# Complexity and variability analyses of motor activity distinguish mood states in bipolar disorder

Petter Jakobsen[1,2]*, Andrea Stautland[2], Michael Alexander Riegler[3], Ulysse Côté-Allard[4], Zahra Sepasdar[5], Tine Nordgreen[6,7], Jim Torresen[4], Ole Bernt Fasmer[1,2], Ketil Joachim Oedegaard[1,2]

**1** NORMENT, Division of Psychiatry, Haukeland University Hospital, Bergen, Norway, **2** Department of Clinical Medicine, University of Bergen, Bergen, Norway, **3** Simula Metropolitan Center for Digitalisation, Oslo, Norway, **4** Department of Informatics, University of Oslo, Oslo, Norway, **5** School of Electrical and Computer Engineering, Shiraz University, Shiraz, Iran, **6** Division of Psychiatry, Haukeland University Hospital, Bergen, Norway, **7** Department of Global Public Health and Primary Care, University of Bergen, Bergen, Norway

* petter.jakobsen@helse-bergen.no

**Data Availability Statement:** This study involves human research participant data that cannot be public shared at this point in time due to ethical restrictions. The analyzed data are part of an

## Abstract

Changes in motor activity are core symptoms of mood episodes in bipolar disorder. The manic state is characterized by increased variance, augmented complexity and irregular circadian rhythmicity when compared to healthy controls. No previous studies have compared mania to euthymia intra-individually in motor activity. The aim of this study was to characterize differences in motor activity when comparing manic patients to their euthymic selves. Motor activity was collected from 16 bipolar inpatients in mania and remission. 24-h recordings and 2-h time series in the morning and evening were analyzed for mean activity, variability and complexity. Lastly, the recordings were analyzed with the similarity graph algorithm and graph theory concepts such as edges, bridges, connected components and cliques. The similarity graph measures fluctuations in activity reasonably comparable to both variability and complexity measures. However, direct comparisons are difficult as most graph measures reveal variability in constricted time windows. Compared to sample entropy, the similarity graph is less sensitive to outliers. The little-understood estimate Bridges is possibly revealing underlying dynamics in the time series. When compared to euthymia, over the duration of approximately one circadian cycle, the manic state presented reduced variability, displayed by decreased standard deviation ($p = 0.013$) and augmented complexity shown by increased sample entropy ($p = 0.025$). During mania there were also fewer edges ($p = 0.039$) and more bridges ($p = 0.026$). Similar significant changes in variability and complexity were observed in the 2-h morning and evening sequences, mainly in the estimates of the similarity graph algorithm. Finally, augmented complexity was present in morning samples during mania, displayed by increased sample entropy ($p = 0.015$). In conclusion, the motor activity of mania is characterized by altered complexity and variability when compared within-subject to euthymia.

ongoing study where the sensitive patient information has not yet been de-identified. The current study (2017/937) will end no later than November 15, 2025, and all data will them be de-identified, as stated by the University Hospital ethical committee. Contact information for the ethical committee: The Regional Ethics Committee, REK vest, University of Bergen, Faculty of Medicine, Postboks 7804, 5020 Bergen, Norway (rek-vest@uib.no).

**Funding:** This work was funded by the Norwegian Research Council (agreement 259293). The funder had no role in study design, data collection and analysis, decision to publish, or preparation of the manuscript.

**Competing interests:** The authors have declared that no competing interests exist.

## Introduction

Change in energy, expressed as either retarded or agitated psychomotor activity, is a core symptom of mood episodes in bipolar disorder [1–3]. Psychomotor activity can be measured using wrist-worn piezoelectric accelerometers, recording acceleration in the three-dimensional space [4]. The foundation for the treatment of bipolar disorder is to avoid future relapses [5]. Patients typically experience changes in sleep and energy in the beginning of new episodes, often without subjectively recognizing these changes [6]. However, if such changes can be identified as they happen, effective interventions can be established to inhibit new mood episodes of a severity requiring hospitalization [5]. The objective information contained in motor activity data has great potential for early detection of emerging mood episodes, improving the management of the disorder and reducing the burden of disease [7–9].

According to systematic reviews [1,2], the bipolar manic state is associated with increased variability and complexity in psychomotor activity patterns when compared to healthy controls. The depressed state is associated with reduced mean motor activity, increased variability and simplicity in activity patterns when compared to healthy controls, and reduced mean activity compared to the manic state. Overall, people with a bipolar disorder diagnosis have reduced mean motor activity compared to healthy controls. Few studies of bipolar manic psychomotor energy have used modern equipment to record motor activity [2–4]. One group [10] found increased variance and reduced mean activity when comparing hospitalized manic patients to healthy controls. A subsequent case series study [11], comparing mood episodes from a single patient, reported elevated activity levels and patterns of amplified complexity in mania compared to depression. Another group [12] reported irregular circadian rhythms in a group of euthymic bipolar patients compared to healthy controls, and attenuated circadian cycles for an essential matching patient group in a manic or mixed state. A similar trend was observed in a study of ecological accelerometer recordings [13], reporting a correlation between manic symptom severity and diminishment of diurnal rhythmicity. Furthermore, a study of circadian rhythmicity in bipolar disorder found no difference in physical activity when comparing hospitalized manic patients to healthy controls and depressed patients [14]. However, patients in a manic episode did wake up significantly earlier compared to when they were in remission, and they had significantly poorer sleep quality when manic. These previous studies were all group wise comparisons or had few participants, substantiating the need for more studies on motor activity and circadian rhythms in bipolar patients. Especially for studies of change in motor activity related to change in mood state, a within subject design, where subjects are their own controls, are in demand [15].

Disrupted circadian rhythms are characteristic symptoms of mood episodes in bipolar disorder [16,17], and disturbed sleep-wake cycles are typical symptoms of mood episodes [18]. The circadian system is best described as a complex system of recurring interlocked rhythms, mainly harmonized by the suprachiasmatic nucleus in the anterior hypothalamus, but also cued by hormones and adjusted by external synchronizers such as light exposure and social life patterns [19]. Interlocked with the 24-h circadian rhythm is a 4-hour ultradian clock, which regulates rest-activity patterns [20]. Increased dopamine function results in a disturbed cyclical clock out of sync with the circadian rhythm, and is associated with manic symptoms [21]. Increased dopamine levels are also associated with arousal of the behavioral activation system [22], a system associated with increased goal directed activity triggering energy and euphoria. Evidence suggests mania is linked to a hypersensitivity in the behavioral activation system [23]. Consequently, motor activity recordings register the complex dynamic interplay of circadian and ultradian biological cycles in interaction with social rhythms.

There is no general standardized method for analyzing accelerometer data [4], nonetheless, non-linear dynamic analyses are considered the most useful method to sufficiently disclose the

information contained in motor activity [24]. Simple linear models are found to be incapable of revealing the variability and complexity characterizing the activity patterns of bipolar disorder [2,10,11]. Recently, the similarity graph algorithm, a method based on evaluating patterns of compounds in time series, has revealed a promising ability to discriminate between diagnostic groups in motor activity recordings. The method has successfully differentiated between depression, schizophrenia and healthy controls [25], as well as patients with ADHD from clinical controls (depression/anxiety) and healthy controls [26].

The aim of this study was to characterize motor activity patterns of the bipolar manic state by comparing manic patients intra-individually to their euthymic selves, applying common linear and non-linear mathematical models, as well as the similarity graph algorithm.

## Materials and methods

### Participants

The participants eligible for this experiment (n = 16) were patients admitted to Haukeland University Hospital, Bergen, Norway, diagnosed with a bipolar disorder according to ICD-10, and in an ongoing manic episode (ICD-10 diagnosis F31.1 and F31.2; current episode manic without/with psychotic symptoms). The clinical psychiatrists residing at the hospitals' two closed wards for affective disorders suggested potential candidates. Patients considered unable to consent by the referring psychiatrist were not invited to participate. Inclusion criteria were Norwegian speaking individuals between 18 and 70 years diagnosed with bipolar disorder, able to comply with instructions and with an IQ clinically evaluated to be above 70. Exclusion criteria were previous head trauma needing hospital treatment, having an organic brain disorder, substance dependence (excluding nicotine), or being in a withdrawal state (see Table 1 for participant details). The study protocol was approved by The Norwegian Regional Medical Research Ethics Committee West (2017/937). Informed, written consent was obtained from all participants, and no compensations for participating in the study were given.

### Clinical assessments

Patient mood state was evaluated at two assessments points by the Young Mania Rating Scale (YMRS) [27]. YMRS rates the severity of mania based on clinical observation of the patients, as well as the patients' subjective description of their clinical condition during the past 48 hours. The total score spans from 0 to 60, and YMRS scores below 10 is considered as being in remission, or in a euthymic state [28]. The severity of depressive symptoms were rated on the Montgomery Asberg Depression Rating Scale (MADRS) [29]. Diagnosis was validated at the second assessment point by research personnel trained in the use of the Norwegian translation of the Mini International Neuropsychiatric Interview (MINI) version 6.0.0 [30].

### Recordings of motor activity

Motor activity was recorded for 24 hours using the Empatica E4 wristband containing several integrated sensors [31], worn on the participants' dominant hand [32]. The participants were assessed twice, first at inclusion and later in remission, at discharge from the hospital or after hospitalization.

The 3-axis accelerometer module integrated within the wristband measured acceleration in gravitational force equivalents (g), with a detection sensitivity of 0.0156 g, and a sampling frequency of 32 Hz. The raw data files were processed in RStudio version 1.2.1335. The absolute mean of the 3-axis' activity counts per minute was calculated for each time series of motor activity, by the formula | SQRT ($x2 + y2 + z2$)–Gravity|, then 1920 lines (the sum of 32 Hz multiplied

**Table 1. Patients characteristics and demographics (N = 16).**

| | |
|---|---|
| Mean age (SD) | 44 (12) |
| Range age (minimum—maximum) | 21–65 |
| Sex (male / female) | 8 / 8 |
| Marital status: | |
| Single / Divorced (%) | 56 |
| Married / Cohabiting (%) | 44 |
| Employment status: | |
| Employed /Student (%) | 37 |
| Unemployed (%) | 19 |
| Disability benefit /Retired (%) | 44 |
| Highest level of education completed: | |
| Junior high school (%) | 12 |
| High school / Vocational studies (%) | 31 |
| University / higher education (%) | 57 |
| Mean age at first hypomanic/manic episode (SD) | 26 (11) |
| Mean age at first depressive episode (SD) | 26 (13) |
| Psychotic symptoms in mood episodes, lifetime (%) | 81 |
| Manic episode (No psychosis[a] / Psychosis[b]) | 7 / 9 |
| YMRS manic episode, mean (SD) | 22 (6)* |
| YMRS when in remission, mean (SD) | 3 (2)* |
| MADRS manic episode, mean (SD) | 6 (4) |
| MADRS when in remission, mean (SD) | 5 (5) |
| Percent activity recorded in summer (manic/euthymic)[c] | 44 / 38 |
| Psychopharmacological treatment (n): | |
| Mood Stabilizers: | |
| Lithium (manic / euthymic) | 5 / 6 |
| Valproate (manic / euthymic) | 7 / 6 |
| Lamotrigine (manic / euthymic) | 2 / 3 |
| Antipsychotics (manic / euthymic) | 14 / 13 |
| Antidepressant (manic / euthymic) | 2 / 2 |
| Benzodiazepines (manic / euthymic) | 5 / 1 |

Abbreviations: SD = standard deviation.

[a] ICD-10 diagnosis: F31.1, current episode manic without psychotic symptoms.

[b] ICD-10 diagnosis: F31.2, current episode manic with psychotic symptoms.

* Mania vs euthymia–YMRS significantly different (p < 0.001), Paired Samples t-test.

[c] Summer defined as the half-year period between the vernal and autumnal equinoxes.

by 60 seconds) were summed and divided by 1920. The calculated outputs are comparable to the motor activity data analyzed in previous studies of bipolar disorder [2,3,11,15,25].

The devices recorded motor activity for an average of 1535 (220) (mean (standard deviation)) minutes, range 1190 to 2067 minutes. As all sequences need to be of similar length for the similarity graph approach, 1190 minutes was defined as the time series length to be analyzed. The average starting time for the recordings was around midday (13:04 (1:25), range 09:52 to 15:52). There were no significant differences (t-test) in starting time, and for that reason, the first 1190 minutes were used from all recordings. A threshold of less than 5% missing data in the specific time series was considered acceptable [33], and missing values were replaced with the mean of the relevant time series. Two participants were excluded from the analysis due to

missing data, attributed to removing the sensor-wristband before sleep. A visual presentation of all included motor activity time series are available in the supporting information, with the manic recordings presented in S1 Fig. and the euthymic recordings in S2 Fig.

Two shorter 120-minute periods, in the morning and evening, were analyzed in addition to the 1190 minutes, Time series time-points were decided post-hoc, upon inspection of available data for each subject. Initially, a fixed tentative morning period was planned between 08.00 and 10.00, as well as an evening period between 20:00 and 22:00. As the mean start of daytime activity was 7:49 (01:48), the criteria for proposing the specific time series were; being 12 hours apart and least likely to be biased by circadian sleep-wake cycles. To maximize the amount of included data from the participants, with minimal missing data in the time series, each time series was adjusted according to visually observed activation patterns in the motor activity data, like late awakening and early night sleep. This was to avoid both individual and mood state related sleep patterns affecting the results. Consequentially, the mean start time for the 2-hour morning epoch and end time of the evening epoch were 08:30 (0:54) and 21:42 (1:46), respectively. One participant was not included in the morning and evening analyses, as the participants' motor activity recording terminated less than two hours after end of sleep, which led to an exceedance of the acceptable missing data limit. The key information of the analyzed motor activity files is available in the supportive S1 Table.

## Mathematical analyses

The motor activity time series were analyzed for mean activity counts per minute. Two estimates of variance, expressing the stability of the mean in a time series, were calculated for the mathematical analyses. Standard deviation (SD) is a measure of how dispersed the data are in relation to the mean. Low standard deviation means data are clustered around the mean, and high standard deviation indicates data are more spread out. The standard deviation is calculated as the square root of variance by determining each data point's deviation relative to the mean. The coefficient of variation (CV) is obtained by dividing SD to the Mean. It describes the variability of a sample relative to its mean. This measure is unitless, expressed as a percentage, and recommended applied in time series with unstable means [34]. In our experience, this definitely applies to time series of motor activity. Therefore, in this paper, SD is in fact CV. The root mean square successive differences (RMSSD) is the root mean square of successive differences between all the time epochs, and indicates how much a set of data varies within itself [35]. For the same reason as for SD, RMSSD is given as a ratio to the mean. Finally, the RMSSD/SD ratio was calculated. Because RMSSD provides the variability between successive intervals instead of solely variability, as assessed by SD, the RMSSD / SD ratio reveal how scattered the data are in themselves.

An autocorrelation function is a mathematical tool for identifying repeated patterns in time series by determining the degree of relationship between the time series and an offset copy [36]. The autocorrelation at lag 1 is the correlation of a time series with itself delayed one interval, and is a common method applied within dynamic system research [37].

Sample entropy is a nonlinear index of complexity in dynamic time series. Higher values indicate intricacy and randomness in patterns, while smaller values point toward predictability and regularity. Sample entropy is defined as the negative natural logarithm of the likelihood of a pointwise matching sequence (m) within a certain tolerance (r) matching the next point [38]. Based on previous studies on nonlinear analysis of motor activity, the following values were selected: m = 2 and r = 0.2 standard deviation [10].

The Symbolic Dynamics method [39] also gives an indication of the complexity of the time series [10,40]. The principle of the method is to transform time series into strings consisting of

numbers between 1 and 6, based on dividing the activity counts into six equal segments, where maximum and minimum values are limited to mean plus-minus 3 SD to counteract the effect of outliers. Finally, the number string is divided into overlapping sequences with three consecutive numbers, giving 216 possibilities for different patterns.

**The similarity graph algorithm.** Before presentation of the similarity graph-based method applied in this study, some basic definitions of graph theory need to be defined. Graphs are Mathematical structures which are used to model the relations between objects. A graph G is an ordered pair (V, E), where V is the set of nodes and E is the set of edges of G. The ends of an edge are said to be incident with the edge. Two nodes which are incident with a common edge are neighbors. An induced subgraph of a graph is a graph formed from a subset of the nodes of G and all the edges connecting pairs of nodes in that subset. A connected component of G is an induced subgraph H which is not a proper subgraph of a connected subgraph of G. Let $e$ be an edge of G. If G-$e$ has more connected components than G, then $e$ is a bridge. A complete graph is a graph in which any two nodes are connected an edge. A complete subgraph of G with $k$ nodes is called a $k$-clique of G.

In this paper, we apply the nonlinear similarity graph algorithm which is based on work done by Lacasa et al. [41], and has been comprehensively described [25,26]. This algorithm transforms a time series S = ($x_1$, $x_2$, . . . $x_n$) into an undirected similarity graph G. Each element of time series S corresponds to a node u in V = {1, 2, . . . n} and each node u is assigned a weight equal to the value of $x_u$. The distance between two nodes u and v is |u-v| and when the distance is 1, the two nodes u and v are defined as direct neighbors. Two arbitrary nodes u and v are connected by an edge in G if and only if their distance is below a certain threshold $k$ and max ($x_u$, $x_v$) / min ($x_u$, $x_v$) < 1.2. Clearly, by changing the values of $k$, different similarity graphs are obtained. Defining 20% as the threshold for similarity is founded in previous studies of motor activity with the sample entropy method [10,11], as well as applied in our groups' previous studies of motor activity and the similarity graph algorithm [25,26]. We used a selection of similarity graph parameters to analyze our data. In summary, any node of the graph with few or no neighbors indicates an alteration in activity. Connected components of the graph indicate substantial shifts in the activity. Bridges expose more subtle activity fluctuations [26]. The number of cliques represents the smoothness of activity fluctuations in a time series [42]. We report on 3-cliques, calculated by a method developed by Chiba & Nishizeki [43]. An illustration of the principles of the similarity graph algorithm presents in Fig 1.

We have calculated the following measures for various distances of $k$: the mean number of edges, the summed number of bridges, components, missing edges between direct neighbors, time points without edges and 3-cliques. We employed $k = 2$, $k = 5$ and $k = 40$ to the 1190 minutes series, and $k = 2$ and $k = 5$ for the 120 minutes series.

## Statistics

Tests of significance were performed in SPSS version 26.0. Paired-Samples T-tests were generally applied, except for the 3-cliques, which were tested using the Related-Samples Wilcoxon Signed Rank Test. A p-value < 0.05 was considered statistically significant when comparing mood states in Tables 2 and 3. When comparing mania and euthymia within both state and subject in the supportive S2 Table, we adjusted the p-value according to a Bonferroni correction for multiple comparisons to avoid a type 1 error [44]. For these analyses, a p-value less than 0.0125 was considered statistically significant. A bivariate Pearson's Correlation test [45] was performed on the manic morning results to examine potential correlations between variance estimates, autocorrelation, complexity and the various outputs of the similarity graph analysis.

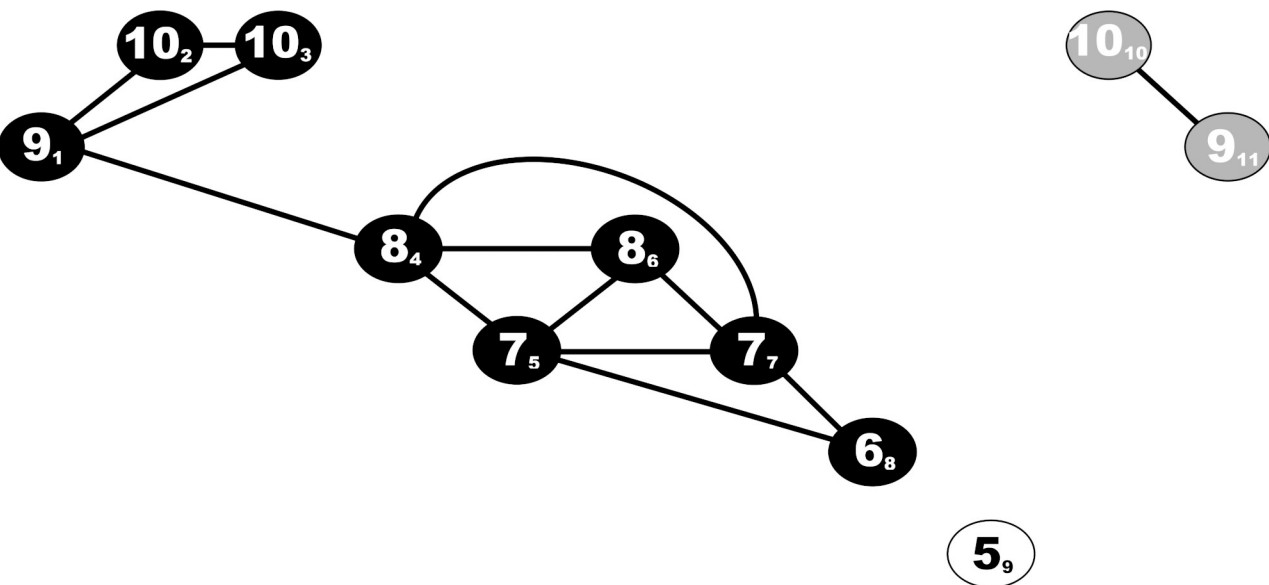

**Fig 1. The similarity graph algorithm exemplified and explained within a $k = 5$ time series.** In this example the similarity graph algorithm transforms a time series S = (9,10,10,8,7,8,7,6,5,10,9) into a graph G, where each element of time series S corresponds to a node in V = {1,2,3,4,5,6,7,8,9,10,11}. The corresponding elements of S and nodes in V are identified as $S_V$ in the figure. Two random nodes u and v are connected by an edge in G if and only if their distance is below a certain threshold $k$ ($|u\text{-}v| < k$), and the ratio of the element values below a threshold defined as max $(x_u, x_v)$ / min $(x_u, x_v) < 1.2$. In this example $k = 5$, and edges are drawn as solid lines in the illustration. The output of the time series are 13 edges, three components (black/white/grey) two bridges ($9_1$–$8_4$, $10_{10}$–$9_{11}$), three missing edges between direct neighbors ($10_3$–$8_4$, $6_8$–$5_9$ and $5_9$–$10_{10}$), one time point without edges ($5_9$), and six 3-cliques ($9_1$–$10_2$–$10_3$), ($8_4$–$7_5$–$8_6$), ($8_4$–$7_5$–$7_7$), ($7_5$–$6_8$–$7_7$), ($8_4$–$8_6$–$7_7$), ($8_6$,$7_7$,$7_5$).

## Results

Forty-five patients hospitalized for a manic episode were invited to participate in the study, of which 34 signed the consent and wore the sensor wristband once for 24 hours. Eighteen of the included patients repeated the recording when in remission. Two of the 18 had one of the assessments inadequately recorded. As a result, 16 patients were sufficiently recorded with the multi-sensor wristband twice. Participant characteristics and demographics are presented in Table 1.

All participants were on medications. Five participants used lithium in combination with one other drug: one with a benzodiazepine, two with an antipsychotic, one with an antidepressant and the final one with Valproate. One of the participants discontinued antipsychotics when euthymic, but the remaining four used the same combination of medications at both measuring points. Eight participants used a mood stabilizing medication other than lithium: six used Valproate combined with antipsychotics during hospitalization, and four when in remission. Of the other two, one participant was started on lithium in addition to valproate and antipsychotics when discharged from the hospital, and the other was switched to Lamotrigine combined with an antipsychotic when in remission. Two participants used Lamotrigine at both assessment points, one in combination with antipsychotics. Finally, two participants used solitary antipsychotics and one participant used a combination of an antipsychotic, antidepressant and benzodiazepine at both assessment points. All antipsychotics prescribed for the manic participants were antidopaminergic (Quetiapine, Olanzapine, Risperidone, Aripiprazole and Zuclopenthixol). Fig 2 shows an example of 24-hour motor activity recordings obtained from one patient when manic (A) and euthymic (B).

Analysis of the 1190-minute recordings of all participants showed no significant differences in mean activity counts per minute between mania and euthymia (Table 2). During mania, SD

**Table 2. Manic and euthymic states compared within subject (N = 14) in 1190 minutes time series of motor activity recordings.**

| | Mania | Euthymia | p |
|---|---|---|---|
| **Mean** | 270.7 (60.1) | 246.3 (35.9) | NS |
| **SD (% of mean)** | 96.4 (21.0) | 116.4 (24.7) | **0.013*** |
| **RMSSD (% of mean)** | 69.8 (14.7) | 70.0 (12.3) | NS |
| **RMSSD / SD** | 0.73 (0.08) | 0.62 (0.14) | **0.037*** |
| **Symbol Dynamics** | 130 (16) | 115 (21) | NS |
| **Sample Entropy** [§] | 0.37 (0.14) | 0.27 (0.10) | **0.025*** |
| **Autocorrelation lag 1** | 0.73 (0.06) | 0.80 (0.09) | NS |
| **Edges ($k$ = 2)** | 1.93 (0.31) | 2.23 (0.29) | **0.039*** |
| **Components ($k$ = 2)** | 448 (82) | 380 (73) | NS |
| **Bridges ($k$ = 2)** | 257 (40) | 210 (38) | **0.012*** |
| **Missing edges ($k$ = 2)** | 578 (95) | 485 (88) | **0.039*** |
| **Points no edges ($k$ = 2)** | 268 (67) | 233 (53) | NS |
| **3-Cliques ($k$ = 2)** | 398 (103) | 509 (102) | **0.035**** |
| **Edges ($k$ = 5)** | 4.24 (0.75) | 4.88 (0.73) | NS |
| **Components ($k$ = 5)** | 258 (56) | 240 (46) | NS |
| **Bridges ($k$ = 5)** | 194 (42) | 148 (40) | **0.026*** |
| **Missing edges ($k$ = 5)** | 581 (95) | 488 (87) | **0.039*** |
| **Points no edges ($k$ = 5)** | 132 (37) | 127 (29) | NS |
| **3-Cliques ($k$ = 5)** | 3253 (964) | 4133 (966) | **0.041**** |
| **Edges ($k$ = 40)** | 20.80 (5.28) | 22.98 (4.84) | NS |
| **Components ($k$ = 40)** | 79 (9) | 90 (10) | **0.005*** |
| **Bridges ($k$ = 40)** | 51 (7) | 47 (8) | NS |
| **Missing edges ($k$ = 40)** | 615 (92) | 521 (87) | **0.030*** |
| **Points no edges ($k$ = 40)** | 93 (5) | 98 (5) | **0.021*** |
| **3-Cliques ($k$ = 40)** | 94072 (51195) | 115615 (49183) | NS |

All results are given as mean (standard deviation).

Abbreviations: SD = standard deviation, RMSSD = root mean square successive difference, NS = not significant.

[a] Sample Entropy: m = 2, r = 0.2

* Significant at a $p < 0.05$ level, Paired Samples t-test.

** Significant at a $p < 0.05$ level, Related-Samples Wilcoxon Signed Rank Test.

was significantly reduced and the RMSSD/SD ratio was significantly increased. Furthermore, the participants had significantly higher sample entropy values when manic. The similarity graph algorithm yielded statistically significant differences for several parameters. For the $k$ = 2 distance the manic state exhibited reduced occurrence of edges and an increased number of bridges and missing edges compared to the euthymic state. The number of 3-cliques was also significantly reduced during mania, regarding both $k$ = 2 and $k$ = 5 distances. For the latter, mania was associated with an increased number of bridges and missing edges compared to euthymia. The $k$ = 40 neighborhood analysis revealed significantly less components, more missing edges and less time points without edges in mania compared to euthymia (Table 2).

The analysis of the 120-minute morning recordings revealed no significant differences for mean activity, variability or autocorrelation between mood states (Table 3). The complexity estimates sample entropy and symbol dynamics were significantly augmented for mania. Most parameters of the similarity graph algorithm differed significantly in the $k$ = 2 neighbors analysis. We found a reduction in edges and 3-cliques, as well as increased numbers of components, time points without edges and missing edges between direct neighbors, in the manic compared

**Table 3. Manic and euthymic states compared within subject (N = 15) in 120 minutes time series of motor activity.**

| | Morning | | | Evening | | |
|---|---|---|---|---|---|---|
| | **Mania** | **Euthymia** | **p** | **Mania** | **Euthymia** | **p** |
| **Mean** | 411.1 (110.0) | 341.6 (117.0) | NS | 377.1 (128.5) | 311.0 (91.4) | NS |
| **SD (% mean)** | 70.8 (15.9) | 77.7 (24.8) | NS | 76.0 (18.8) | 85.8 (24.2) | NS |
| **RMSSD (% mean)** | 64.8 (13.3) | 66.7 (37.5) | NS | 68.8 (20.7) | 67.6 (26.3) | NS |
| **RMSSD / SD** | 0.93 (0.15) | 0.84 (0.21) | NS | 0.90 (0.12) | 0.79 (0.18) | NS |
| **Symbol Dynamics** | 50 (7) | 43 (8) | **0.014**[*] | 47 (9) | 41 (14) | NS |
| **Sample Entropy** [a] | 1.14 (0.36) | 0.78 (0.38) | **0.015**[*] | 1.01 (0.34) | 0.78 (0.44) | NS |
| **Autocorrelation lag 1** | 0.56 (0.14) | 0.62 (0.18) | NS | 0.58 (0.10) | 0.67 (0.14) | NS |
| **Edges ($k = 2$)** | 1.15 (0.18) | 1.49 (0.34) | **0.007**[*] | 1.24 (0.36) | 1.53 (0.44) | **0.032**[*] |
| **Components ($k = 2$)** | 65 (7) | 55 (11) | **0.023**[*] | 63 (13) | 55 (11) | **0.018**[*] |
| **Bridges ($k = 2$)** | 35 (7) | 33 (10) | NS | 32 (6) | 33 (10) | NS |
| **Missing edges ($k = 2$)** | 82 (6) | 72 (10) | **0.006**[*] | 78 (11) | 71 (14) | NS |
| **Points no edges ($k = 2$)** | 43 (7) | 34 (9) | **0.021**[*] | 41 (12) | 34 (8) | **0.036**[*] |
| **3-Cliques ($k = 2$)** | 11 (5) | 21 (11) | **0.009**[**] | 15 (9) | 22 (15) | NS |
| **Edges ($k = 5$)** | 2.56 (0.44) | 3.28 (0.85) | **0.031**[*] | 2.62 (0.78) | 3.23 (1.07) | **0.047**[*] |
| **Components ($k = 5$)** | 40 (9) | 36 (9) | NS | 43 (10) | 38 (7) | NS |
| **Bridges ($k = 5$)** | 26 (5) | 22 (9) | NS | 24 (8) | 20 (6) | **0.020**[*] |
| **Missing edges ($k = 5$)** | 83 (5) | 74 (9) | **0.009**[*] | 81 (11) | 74 (14) | NS |
| **Points no edges ($k = 5$)** | 26 (6) | 24 (6) | NS | 27 (7) | 24 (5) | NS |
| **3-Cliques ($k = 5$)** | 83 (32) | 160 (89) | **0.012**[**] | 98 (62) | 166 (132) | NS |

All results are given as mean (standard deviation).

Abbreviations: SD = standard deviation, RMSSD = root mean square successive difference, NS = not significant.

[a] Sample Entropy: m = 2, r = 0.2.

[*] Significant at a $p < 0.05$ level, Paired Samples t-test.

[**] Significant at a $p < 0.05$ level, Related-Samples Wilcoxon Signed Rank Test.

to the euthymic morning recordings. The number of bridges did not differ significantly between mood states. The $k = 5$ distance analysis revealed a reduced number of edges and 3-cliques, as well as an increased number of missing edges between direct neighbors in mania compared to euthymia. Similarly to the morning analyses, the 120-minute evening recordings showed no significant differences between mood states for mean activity, variability, autocorrelation or complexity. Several parameters of the similarity graph analyses presented statistically significant differences between the manic and euthymic states. We found a reduced number of edges in mania for both $k = 2$ and $k = 5$ analyses. We also found a significantly increased number of components and time points without edges for the $k = 2$ distance, and a significantly increased number of bridges for the $k = 5$ distance.

When comparing the 120-minute morning and evening periods within mood states, we found no significant differences for mania or euthymia. The results are presented in the supportive S2 Table.

Table 4 presents the relationships between the various $k = 2$ similarity graph estimates and the estimates of variance, complexity and autocorrelation. The two estimates of variance (SD and RMSSD) are strongly correlated with each other, as well as negatively correlated with the similarity graph estimate bridges. Furthermore, SD is strongly negatively correlated with sample entropy, which in turn is moderately negatively correlated with both 3-cliques and autocorrelation. The RMSSD/SD estimate presents near perfect negative correlation with autocorrelation, and is moderately correlated with sample entropy. Number of edges is either

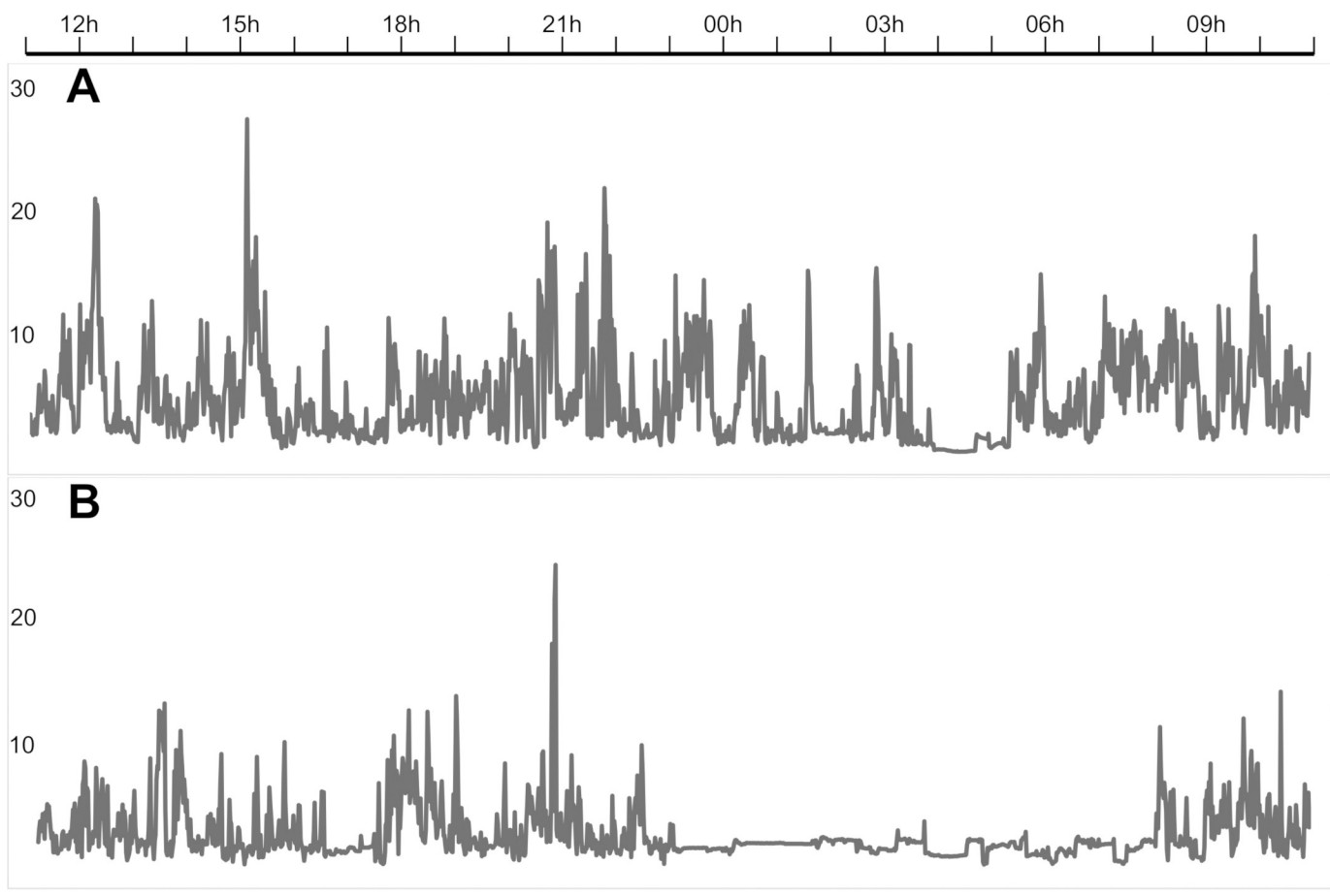

**Fig 2. 24-hour accelerometer recordings from a study participant.** The patient was recorded during mania when hospitalized (A), and later in remission (B). The figure shows the activity counts (gravitational force equivalents) per minute over 24 hours, from 11 a.m. to 11 a.m. the next day.

strongly positively or negatively correlated with all other estimates of the similarity graph, except for bridges. Likewise, the number of connected components is strongly correlated with the other similarity graph estimates, except for a moderately negative correlation with both 3-cliques and bridges.

## Discussion

The novelty of this study is that our findings provide empirical support of how the agitated manic psycho motoric activity differs from euthymia intra-individually in humans with bipolar disorder. In the first place, the clinical relevance of this study may appear to be of minimal importance; however, as an evidence-based foundation for future innovations in clinical aids, the clinical relevance of this study may be of significant importance.

Our results suggest that the bipolar manic state is associated with distinct deviating motor activity when compared within subjects to their euthymic selves. Comparing the manic state to euthymia in the 20-hour (1190 minutes) time series revealed that mania is characterized by reduced variability, displayed by decreased standard deviation, and increased complexity, displayed by augmented sample entropy. We also discovered increased complexity during mania in the 120-minute morning sequences, displayed by significantly augmented values for both sample entropy and symbol dynamics. The similarity graph algorithm $k = 2$ distance appears

**Table 4. Pearson correlation analyses for manic morning variables (n = 15).**

| | SD | RMSSD | RMSSD /SD | Autocor- relation | Sample Entropy | Symbol Dynamics | Edges [a] | Comp. [a] | Bridges [a] | Missing Edges [a] | No Edges [a] |
|---|---|---|---|---|---|---|---|---|---|---|---|
| RMSSD | **0.782**\*\* | | | | | | | | | | |
| RMSSD/SD | -0.407 | 0.244 | | | | | | | | | |
| Autocorrelation | 0.429 | -0.212 | **-0.996**\*\* | | | | | | | | |
| Sample Entropy | **-0.752**\*\* | -0.475 | **0.515**\* | **-0.551**\* | | | | | | | |
| Symbol Dynamics | -0.457 | -0.369 | 0.183 | -0.169 | 0.324 | | | | | | |
| Edges [a] | -0.065 | -0.113 | -0.098 | 0.086 | -0.405 | -0.323 | | | | | |
| Components [a] | 0.291 | 0.374 | 0.120 | -0.096 | 0.191 | 0.311 | **-0.933**\*\* | | | | |
| Bridges [a] | **-0.675**\*\* | **-0.755**\*\* | -0.049 | 0.021 | 0.392 | 0.079 | 0.237 | **-0.556**\* | | | |
| Missing Edges [a] | 0.255 | 0.346 | 0.154 | -0.138 | 0.275 | 0.165 | **-0.883**\*\* | **0.859**\*\* | -0.285 | | |
| Points No Edges [a] | 0.500 | **0.615**\* | 0.135 | -0.100 | -0.021 | 0.221 | **-0.773**\*\* | **0.920**\*\* | **-0.688**\*\* | **0.801**\*\* | |
| 3-Cliques [a] | 0.185 | 0.231 | 0.012 | -0.002 | **-0.560**\* | -0.103 | **0.847**\*\* | **-0.615**\* | -0.211 | **-0.707**\*\* | -0.369 |

Abbreviations: SD = standard derivation, RMSSD = root mean square successive differences.

[a] $k = 2$.

\*\* Correlation is significant at the 0.01 level (2-tailed).

\* Correlation is significant at the 0.05 level (2-tailed).

to possess the strongest discriminating abilities, and we found significant differences between mania and euthymia for this distance in all three sequences. The manic state was associated with less edges and an increased number of missing edges between direct neighbors. This indicates increased shifts in activity, causing fewer allied time points due to non-similarity within the area limits. Nonetheless, the increased number of bridges for the manic state in the 20-hour time series indicates a certain degree of smoothness in the activity shifts, as the time points are to some extent connected with at least one edge. Furthermore, both 120-minute sequences revealed significant changes in the connected component estimates, indicating more roughness in the manic morning and evening motor activity, due to missing edges caused by non-similarity between time points. Finally, for the morning and 20-hour sequences, the high number of 3-cliques for euthymia indicates more stable and robust motor activity patterns compared to the manic activity.

When changing the values of $k$, different similarity graphs are obtained, and different results are to be expected [26]. However, the results of the $k = 5$ neighborhood analyses were to some extent similar to the $k = 2$ results for both the 20-hour and the 120-minute evening sequences. The $k = 5$ morning sequences only resulted in significant differences in edges, missing edges and 3-cliques. This was somewhat unexpected, as the morning epoch presented significant differences for most of the $k = 2$ measures. On the other hand, these deviating results may be a revelation of subtle fluctuations in the morning activity patterns, undetectable when increasing the value of $k$ [26]. In the evening series the quantity of bridges were significantly increased solely for the $k = 5$ graph, while the number of components were increased solely for the $k = 2$ graph. This indicates that the similarity graph obtained by increasing values of k, reduces the non-similarity between time points in the series. For the $k = 40$ patterns in the 20-hour sequence, three measures were significantly different between states; mania exhibited increased missing edges, reduced number of components and fewer time points without connected edges compared to euthymia. In other words, mania seems to be associated with unstable activity patterns, due to increased missing edges, while euthymia seems to be associated

with more dramatic shifts in activity, due to a higher number of components and unconnected time points.

It is well established that variability and complexity analyses are needed to adequately reveal the information contained in motor activity data [1,2,11]. However, our results suggest that the similarity graph is a more sensitive and finely calibrated tool for such a task. This is evident as the variability estimates were mostly insignificantly altered when comparing mood states for the 120-minute time series, while the similarity graph revealed significant differences in all sequences, independent of the area limits defined by various sizes of $k$ and length of time series. This also applies to the complexity measures, although both sample entropy and symbol dynamics were significantly altered in the morning sequence.

The similarity graph estimates can be regarded as a combination of variability and complexity measures. The patterns of connections and missing connections express fluctuations in activity similar to other estimates of variance, like SD and RMSSD [26]. However, direct comparisons are difficult, as SD reveals the variability of the entire time series, while graph measures like edges, components and 3-clicks reveal variability in constricted time windows. In contrast to the other graph estimates, missing edges (between direct neighbors) are calculated over the complete time series. It provides information quite comparable to RMSSD [35], as both estimates expresses the relationship between sequential points in the whole time series. Nevertheless, the relationship between variance and graph estimates appears to be unrelated, as no correlations are indicated in Table 4, except for bridges. The estimate of bridges exposes more subtle activity fluctuations and is not fully understood, but suspected to reveal underlying dynamics of the time series [26]. The distinctiveness of bridges is emphasized by the correlation analyses, as it correlates poorly to the other graph estimates, except for a moderate negative relationship to components and time points without edges. These findings elucidate novel characteristics of the enigmatic bridge estimate. Finally, bridges provide further evidence of the similarity graph as a more sensitive and finely calibrated tool than traditional variance estimates, as no significant differences for SD or RMSSD were found in the evening time series, while a statistically significant difference between mood states was observed in the bridges ($k$ = 5) comparison (Table 3).

The patterns of connections and missing connections express the intricacy or simplicity of activity alterations within a shorter time series. This resembles both sample entropy and symbolic dynamics, which are also calculated based on the relationship between time points within a shorter time-series. However, we found no suggestions of a relationship between symbolic dynamics and any of the similarity graph estimates, and sample entropy displayed only a moderate negative correlation with 3-cliques (Table 4). Nonetheless, sample entropy is sensitive to outliers, compared to the similarity graph estimates [26]. Outliers increase the standard deviation, which again increases the probability of two points being valued as similar, and the result comes to be incorrectly reduced complexity. Consequently, as the similarity graph algorithm handles this ceiling effect, fewer points (nodes) evaluate as similar, resulting in an increased ability to separate dissimilarities between divergent time series, as presumably exemplified in our analyses.

We found an increased RMSSD/SD ratio during mania in the 20-hour recordings. This supports findings reported by Krane-Gartiser et al. [10], comparing hospitalized manic patients to healthy controls in 24-hour time series. The group reported similar results from 64-minute morning and evening epochs. We found no such RMSSD/SD-ratio differences in the 120-minutes morning and evening time series. Furthermore, Krane-Gartiser et al. discovered the most significant difference between manic patients and controls in the morning period by using the Autocorrelation lag 1 variable. Despite this variable being linked to variability, no such difference was identified in our results. We did, however, find comparable

differences in both morning and evening time series, although mainly in estimates of the similarity graph algorithm, a method not applied by Krane-Gartiser et al. [10].

Although our results show slightly elevated mean activity levels during mania, we found no significant differences between mood states. This was to be expected, as systematic reviews conclude that mania appears better characterized by increased variability and complexity than increased mean level of activity [2]. Moreover, Krane-Gartiser et al. [10], comparing hospitalized manic patients to healthy controls, found significantly lower mean activity levels for the manic patients, assumed due to the patients being pacified by hospitalization and prescribed antipsychotic medication [10]. Consequently, it is reasonable to assume that the elevated activity level in mania compared to depression observed in the previously mentioned case series study [11], is primarily about the motor retardation associated with depression [1,2].

Another aspect to consider regarding activity levels is the influence and manipulation of the behavioral activation system. Evidence suggests mania is linked to a hypersensitivity in the behavioral activation system, a system associated with increased goal directed activity, and generating energy arousal and euphoria [22]. The internal dopaminergic ultradian oscillator clock is associated with the rhythmic patterns of rest-activity [20] and linked to the behavioral activation system [16,22]. The clock is not controlled by the suprachiasmatic nucleus, but habitually oscillates interlocked with the circadian rhythm. When studying the motor activity of laboratory mice on methamphetamine, increased dopamine levels were found to be associated with prolonged dopaminergic cycles out of sync with the circadian rhythm [20]. Furthermore, increased dopamine levels are associated with both the presence of manic symptoms [21] and to stimulation of the behavioral activation system [22].

Both mood stabilizers, like Lithium and Valproate, as well as anti-dopaminergic antipsychotics inhibit the behavioral activation system [21]. All participants in the current study were on such behavioral activation system taming medications during manic recordings, except for one participant on Lamotrigine monotherapy. Altogether, this implies that one can only speculate about the mean activity levels of an unmedicated manic person prior to hospitalization, when living in a stimulating and rewarding environment. The current results of equal activity levels between mood states should be revisited taking these findings into account. Furthermore, as the second recording of motor activity was conducted when the patient was discharged from the pacifying hospital environment, this may have influenced the recordings and potentially reduced differences between mood states. However, the effect of medications must be considered minimal, as there were minimal changes in prescribed medications between recordings.

When comparing morning and evening recordings within states, we found no significant differences for neither the manic nor the euthymic state. This was somewhat unexpected, as existing evidence suggests disrupted circadian and social rhythms as a characteristic of mood episodes in bipolar disorder [16]. In addition, two studies of the manic states' motor activity suggested an association between the manic state and irregular attenuated circadian rhythmicity [12,13], and a study of biochemical rhythms combined with motor activity found shifted circadian rhythm phases in mania [14]. Furthermore, the hypothesis of bifurcation of biological rhythms [46] claims that the master clock in the circadian system, the suprachiasmatic nucleus, switches from its normal 24-hour cycle to a 12-hour phase in mania. Consequentially, diurnal sampling of melatonin levels in manic bipolar patients resulted in two observed peaks in melatonin secretion, as opposed to the normal single peak [47]. However, comparing stationary morning and evening times is most likely not an appropriate method for investigating circadian cycles, as it does not take into account individual circadian variations, such as dissimilar chronotypes and social rhythms [48]. Yet, as a final contradicting point, according to a systematic review [18] disrupted circadian rhythmicity seems to be a trait marker for bipolar disorder, not a state marker for mania. Although there is lack of conclusive evidence for this

assumption, these findings may provide a plausible explanation for the homogeneity in our results when comparing morning and evening recordings within mood states.

## Limitations

There are some structural weaknesses and limitations to this study that may have restricted the findings. To start, the sample size is rather small, which may affect statistical power. However, the within-subject design reduces this weakness. Neither gender, age nor body mass index were controlled for. This may have biased the result as all three variables have previously been found to impact motor activity in group comparisons [15,26]. However, as our findings are results of state changes within subjects, the influential effect of these variables can likely be considered minor. Another possible weakness is the lack of a control group. However, the within-subject design of the study, where subjects are their own controls, presumably makes this a minor issue. Furthermore, the observed changes between states are considered too large to be due to chance. The selection procedure for the 120-minutes morning and evening time series may be reprehensible. However, collaboration with manic patients can be challenging, and their circadian disturbances are well-documented [18]. The manic recordings presented in the supportive S1 Fig and S1 Table illustrate both the noisiness of the data and disrupted circadian cycles. To avoid our results being biased by individualistic sleep patterns, we followed a similar but stricter approach for the selection of morning and evening periods than Krane-Gartiser and colleagues [10] did previously, when analyzing similar patients. Pragmatically, based on the complexity of the data, our selection procedure for the 120-minutes time series should be considered satisfactory.

Our results may also have been moderated by a seasonal effect. Humans are seasonal beings, and both social rhythms and durations of sleep generally follow a seasonal pattern similar to the annual changes in natural light and length of day [17]. Haukeland University Hospital, Bergen, Norway is located at latitude 60.4, a location associated with substantial seasonal change in natural light and length of day. In the data analyzed in this study, the manic episodes were quite evenly distributed between winter and summer; however, 62 percent of the euthymic recordings were collected during the winter months. Therefore, seasonality may have impacted our findings to some degree. But at the same time, in a recent Norwegian survey [49], merely 20% reported a high degree of seasonal variations in mood and behavior, while approximately 60% reported low impact of seasonality.

Our patient sample is highly educated, even more so than average in the highly educated Norwegian population [50]. Regarding this, a study has investigated educational levels and socio-economic status among 257 Norwegian patients diagnosed with bipolar disorder [51]. They found no relationship between educational level and burden of disease. Moreover, our sample ratios of individuals living alone or on disability pension are comparable to those reported in the study, although a higher percentage of the patients in our sample had a lifetime experience of psychosis. A hypothetical explanation for the skewed educational level in our sample could relate to an association between higher education and an understanding of innovational possibilities within bipolar disorder, as well as a selfless wish to contribute to enhanced scientific understanding of the disease.

We have investigated hospitalized bipolar patients during an ongoing manic episode, confirmed and evaluated using a rating scale (YMRS). However, the YMRS scores could be subject to moderation due to the patients being hospitalized in a minimally rewarding environment and being prescribed antipsychotic and mood stabilizing medications.

Euthymia, or remission, was defined as having an YRMS score below 10. This is a slightly stricter threshold than more commonly used 12 or below [52]. MADRS scores are lightly

elevated for the group at both measuring points, implying marginal depressive symptoms present in the groups [53]. Depressive symptoms in mania can represent dysphoric features, which are commonly present in manic episodes [54]. For clarification, agitated depressions and mixed episodes were not included in the study sample. We find it likely that the elevated MADRS scores in the euthymic group are related to residual symptoms, which are common in euthymic bipolar patients [55], and should not negatively affect the representability of the sample.

Despite the declared limitations, our main findings remain robust and well-grounded; a significant intra-subject difference in complexity and variability measures of motor activity exists between manic and euthymic states in bipolar disorder.

### Future work

Motor activity data possesses an innovative potential for the development of a tool or device for early detection of mood episodes in bipolar disorder. To realize this potential, it is necessary to explore the prospects of automatic real-time monitoring through machine learning [56]. Our research group has previously revealed the promising capabilities of various machine learning techniques using motor activity data collected from depressed patients [15]. Future work will further explore the classification capabilities of advanced machine learning models and the potential of applying automatic, real-time monitoring of motor activity for early detection of mood episodes in bipolar disorder. Of particular interest are graph neural networks and Bayesian neural networks. The former is perfectly suited to explore graph-like data structures and their dependencies [57,58], while the latter can help to understand decisions made by the model by quantifying its uncertainty [59].

Motor activity recordings are usually recorded at a sampling rates around 32 Hz and analyzed in one minute epochs, similar to the approach of this study [4]. It is conceivable that hidden information may disappear when analyzing the motor activity in one-minute epochs. Given this, another possible approach is to feed the machines with the absolute mean of the 3-axis' activity counts per Hz, leaving it up to the algorithms themselves to decide appropriate epoch sizes.

### Conclusion

In the present study we have compared motor activity data collected from hospitalized manic bipolar patients to motor activity data collected from the same individuals when in remission. We have applied commonly used linear and non-linear mathematical models, as well as the similarity graph algorithm. No previous studies have compared mania to euthymia intra-individually using such state-of-the-art accelerometer recordings while applying similar methods. We found that the motor activity patterns of the manic state are associated with altered complexity and variability, when compared to euthymia within subjects. Our findings are robust and comparable to results from previous studies comparing bipolar manic patients to healthy controls, and construct a solid evidence-based foundation for future innovations aiming to improve the management of bipolar disorder and reduce burden of disease.

### Supporting information

**S1 Fig. Visualization of all manic motor activity time series (n = 16).** The figures shows the activity counts (gravitational force equivalents) per minute during the complete recording. (PDF)

**S2 Fig. Visualization of all euthymic motor activity time series (n = 16).** The figures shows the activity counts (gravitational force equivalents) per minute during the complete recording. (PDF)

**S1 Table. Description of all time series included in the 120-minute analyzes.** (DOCX)

**S2 Table. Morning and evening differences.** Mania and euthymia compared within subject (N = 15) and within mood state in 120 minutes time series of motor activity. All results given as mean (standard deviation). Abbreviations: SD = standard deviation, RMSSD = root mean square successive difference. [a] Sample Entropy: m = 2, r = 0.2. [b] Paired Samples t-test, except for 3-Cliques that were tested for significant differences with the Related-Samples Wilcoxon Signed Rank Test. For both tests, the significance level was set as $p < 0.0125$, to adjust for multiple comparisons. (DOCX)

**S1 File.** (CSV)

## Acknowledgments

We acknowledge and thank Christoffer Andreas Bartz-Johannessen for statistical consultation and Erlend Eindride Fasmer for developing the similarity graph algorithm software. This publication is part of the INTROducing Mental health through Adaptive Technology (INTRO-MAT) project.

## Author Contributions

**Conceptualization:** Petter Jakobsen, Ole Bernt Fasmer, Ketil Joachim Oedegaard.

**Data curation:** Petter Jakobsen, Ole Bernt Fasmer.

**Formal analysis:** Petter Jakobsen, Ole Bernt Fasmer.

**Funding acquisition:** Tine Nordgreen, Jim Torresen, Ole Bernt Fasmer.

**Investigation:** Ole Bernt Fasmer, Ketil Joachim Oedegaard.

**Methodology:** Petter Jakobsen, Zahra Sepasdar, Ole Bernt Fasmer, Ketil Joachim Oedegaard.

**Resources:** Tine Nordgreen, Jim Torresen.

**Supervision:** Tine Nordgreen, Jim Torresen, Ole Bernt Fasmer, Ketil Joachim Oedegaard.

**Validation:** Petter Jakobsen, Andrea Stautland, Michael Alexander Riegler, Ulysse Côté-Allard, Zahra Sepasdar, Ole Bernt Fasmer, Ketil Joachim Oedegaard.

**Visualization:** Petter Jakobsen, Andrea Stautland, Michael Alexander Riegler, Ulysse Côté-Allard, Zahra Sepasdar, Ole Bernt Fasmer, Ketil Joachim Oedegaard.

**Writing – original draft:** Petter Jakobsen.

**Writing – review & editing:** Andrea Stautland, Michael Alexander Riegler, Ulysse Côté-Allard, Zahra Sepasdar, Tine Nordgreen, Jim Torresen, Ole Bernt Fasmer, Ketil Joachim Oedegaard.

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
