## [Decision Letter · Decision Letter 0]

31 Aug 2021

PONE-D-21-23669

Complexity and variability analyses of motor activity distinguish mood states in Bipolar Disorder

PLOS ONE

Dear Dr. Jakobsen,

Thank you for submitting your manuscript to PLOS ONE. After careful consideration, we feel that it has merit but does not fully meet PLOS ONE’s publication criteria as it currently stands. Therefore, we invite you to submit a revised version of the manuscript that addresses the points raised during the review process.

We look forward to receiving your revised manuscript.

Kind regards,

Gaetano Valenza, Ph.D.

Academic Editor

PLOS ONE

Journal Requirements:

Reviewers' comments:

Reviewer's Responses to Questions

**Comments to the Author**

1. Is the manuscript technically sound, and do the data support the conclusions?

Reviewer #1: Partly

Reviewer #2: Yes

2. Has the statistical analysis been performed appropriately and rigorously? 

Reviewer #1: Yes

Reviewer #2: Yes

3. Have the authors made all data underlying the findings in their manuscript fully available?

Reviewer #1: Yes

Reviewer #2: Yes

4. Is the manuscript presented in an intelligible fashion and written in standard English?

Reviewer #1: Yes

Reviewer #2: Yes

5. Review Comments to the Author

Reviewer #1: The manuscript compared activity patterns in patients (n=14) with bipolar disorder during between their manic states and euthymia, using various mathematical tools, such as sample entropy, or graph-based method. The significant differences in activity patters were found in some indices. However, the results were inconsistent against the choice of the parameter vales of analytical methods, or time of day. In addition, the manuscript includes some important issues possibly affecting the results.

1) The first recording of motor activity was conducted during hospitalization, while the second was conducted out of the hospital? If so, how do the authors think the effects of different recording situations?

2) The process that one-minute activity counts were derived from tri-axial acceleration data were not described. For example, some filtering was applied to the raw data?, how the 3-dimentional data were converted into one-dimensional data? The definition of “activity” is the most important in those research.

3) The mathematical interpretation of the index, RMSSD/SD, might be provided for a potential reader of the journal.

4) Why was the string number set to six in the symbolic dynamics method?

5) Did the results change when a different number of cliques was selected?

6) Age often affects the activity levels or variance. The author may test the correlation of age with significant behavioral indices.

7) Related above, the reviewer suggests to check the correlation among behavioral indices (e.g. sample entropy vs network properties). This may provide further insight on what kind of behavioral dynamics contributes the results obtained.

8) The results derived from 1190 minutes data were possibly affected by the differences in sleep between mania and euthymia. Sleep effects should be carefully treated.

9) As the authors discussed by themselves, the results of the similarity graph method were inconsistent among different hyper-parameter values, or difference in time of day. This would be one of targets for criticism of this manuscript. Showing the systematic changes of behavioral indices with varying parameter vales may be one possible way to show the robustness of the results. More specifically, for example, calculate 3-cliques values for k=2,3,4,5,…, and then plot the results as a function of k. Similarly, effects of the choice the time window of data could be evaluated (e.g., 7:00-9:00, 8:00-10:00,….)

10) Figure 2: no label and no unit on the y-axis

Reviewer #2: This is an interesting study that will be of interest to the readership of PLOS One and to the field of psychiatry more broadly. The authors identify within subject differences in the complexity of motor activity patterns between mania and euthymic states among individuals with bipolar disorder. Although the sample size is small, realistically it is difficult to collect such data and the authors should be commended for their effort. The study provides empirical support of what is clinically intuitive, using a sound methodology and the approaches taken are somewhat novel in the area. Some of the expression of terms in the manuscript is in my opinion quite awkward and could be improved. I have the following feedback that I believe the authors could integrate to improve the quality of this manuscript.

Abstract

- Line 37-38: the authors state that there were fewer edges and bridges in their analysis. It would be helpful for the reader if the authors provided some context in the abstract for people unfamiliar with this approach. As currently written, it assumes knowledge of this analysis before reading the rest of manuscript. Are edge and bridge frequency measures of complexity analogous to entropy? If so please state, if not please differentiate.

Intro

- Line 54/55: needs citation

- Line 56: in my opinion ‘full-blown’ mood episode is a rather colloquial/unrefined way of describing a severe, clinically confirmed mood episode. The authors might consider revising.

- Line 71: ‘dissolved’ is a strange way to describe circadian dysfunction. I would suggest an attenuated or less robust circadian rhythm instead if that is what the authors mean?

- Line 95/96: what do the authors mean when they state that non-linear dynamic analyses are considered the most rewarding method – could the authors revise the term ‘rewarding’. Do you mean recommended method or most informative method? You could additionally elaborate on why this method is superior.

- General comments on introduction and positioning the study: the authors refer to the previous similar work conducted in this area (e.g. review of the Krane-Gartiser et al studies). However, it is not quite clear what the unique contribution of this study is. The authors suggest that previous studies have been conducted using small sample sizes, but so too is the present study one with a small sample size. The authors state that previous work has focused on groupwise differences and in contrast the current study involves a within-subject/intra-individual design. I believe the authors have missed the opportunity to emphasise how their approach advances the evidence demonstrated previously. Perhaps you might consider highlighting the advantages of intra-individual comparisons between acute and remitted mood states in order to communicate the novelty of your study.

Method and Results

- The standard convention in the participants section would be to state the n of sample. This is done elsewhere in manuscript but should also be in the top-line of the participants section.

- The brand and manufacturer of the actigraph used should be stated (or otherwise the manufacturer of the accelerometer that the researchers mounted to the wristband). Is reference 28 supposed to refer to this? Please make more explicit in methods section.

- Do I understand correctly that the accelerometry was only conducted over the space of one day? This is a major limitation and needs to be acknowledged.

- Were there any bespoke R packages used to analyse these data? If so it would be best practice to acknowledge and reference. Also is a markdown of this R code available anywhere for those who wish to apply a similar approach?

- The authors should provide further justification for selected the morning and evening time windows. Was this decided post-hoc upon inspection of available data (implied in manuscript) or was this an a priori decision based on some inference of the circadian rest-activity pattern? If the latter the authors should be aware that the biological morning and evening is determined by the underlying circadian rhythm and, as this was not assessed, arbitrarily defining morning and evening times for the purpose of comparing circadian timing is not appropriate. Later in the discussion the authors use these findings to argue that there is a reduction in circadian variation which I do not think is appropriate. They looked at a diurnal difference within a two hour period in both the morning and evening… any claims made about the underlying circadian rhythm of patients with mania should be revised and the conclusions tempered.

- Regarding the threshold of <5% missing data considered acceptable – do the authors have a reason for selecting this minimum threshold or a source they can reference for this recommendation?

- Why was autocorrelation with a lag of 1 minute chosen? The Krane-Gartiser study to which the authors refer in discussion used several different autocorrelation lag windows.

- For the general reader who is unfamiliar with the RMSSD measure it might be useful to state that this measures instability between successive intervals rather than solely variability as assessed by SD. Furthermore, it is unclear to me why both are given as a ratio of the mean. Is there an assumption that greater motoric activity is correlated with both greater RMSSD and SD and the ratios are an attempt to deal with this confound? (e.g. deriving a coefficient of variation). Similarly, the reason for using the RMSSD as a ratio to SD could be explained. Could the authors please expand on their methods here. Is this to uncouple epoch-to-epoch instability from over variability?

- The authors go into great detail in explaining the similarity graph algorithm. However, I believe they should highlight what this measure reflects and how it different from variability, instability, and complexity as assessed by the other measures. What is the benefit of this analysis and if similar to the entropy and symbolic dynamics method? Could the authors provide context on why this does not create a redundancy of measures?

- Table 2 correction: for the RMSSD (% of mean), the value should be 69.8 rather than using a comma as a decimal point.

- Several results in Table 2 and Table 3 are reported as significant at P<0.05 yet the methods section describes a Bonferroni revised alpha of 0.0125 being the threshold for statistical significance. Please clarify which is the correct threshold and update the results and discussion accordingly.

Discussion

- Generally, the discussion could be improved by situating the novelty of the authors findings within the theme of how these findings might be clinically translated. Much of the discussion section in which the authors interpret their results recaps the results section.

- The use of circadian variation in lines 437-438 is not appropriate for the reasons I have previously detailed. Caution should be applied in making inferences about the circadian timing system as there is no contemporaneous measure of endogenous clock function. Please give consideration to this point in the revised manuscript.

- Anitpsychotic and mood stabiliser medications are mentioned in the discussion, but these are not highlighted as limitations. Both may have profound effects on motor rest-activity patterns and surely obfuscate the effect of mood state.

- The authors state that no previous studies have applied such state-of-the-art measures. Yet, as I understand it, the novelty is limited to the similarity graph algorithm. Most of the measures of instability/complexity/autocorrelation have previous been applied in other studies. Therefore, the authors should revise this claim.

6. PLOS authors have the option to publish the peer review history of their article (what does this mean?). If published, this will include your full peer review and any attached files.

Reviewer #1: No

Reviewer #2: No

---

## [Author Response · Author response to Decision Letter 0]

4 Oct 2021

Journal Requirements:

Answer: In our opinion, this is the way it should be.

Answer: Two minimal underlying data sets have been uploaded as Supporting Information files, fully anonymized containing no potentially identifying patient information. The file “manic_euthymic_24h_minimal_data_set” contains the values behind all measures reported in table 2. The file “manic_euthymic_morning_evening_minimal_data_set” contains all values behind the measures reported in table 3, 4 and 5.

5. Review Comments to the Author

Reviewer #1: 

The manuscript compared activity patterns in patients (n=14) with bipolar disorder during between their manic states and euthymia, using various mathematical tools, such as sample entropy, or graph-based method. The significant differences in activity patters were found in some indices. However, the results were inconsistent against the choice of the parameter vales of analytical methods, or time of day. In addition, the manuscript includes some important issues possibly affecting the results.

1) The first recording of motor activity was conducted during hospitalization, while the second was conducted out of the hospital? If so, how do the authors think the effects of different recording situations? 

Answer: In the authors opinion the effect of the different recording situations may have lessened the measured differences between mood states, as hospitalization involves being placed in a minimally stimulating and rewarding environment, potentially very dissimilar to the situation out in the community. This is partly discussed in the manuscript, both generally (line 501 – 505 in the revised manuscript w/track changes) and regarding the YMRS score (line 569 – 571). To clarify our opinion, the following text was added to the manuscript “Furthermore, as the second recording of motor activity was conducted when the patient was discharged from the pacifying hospital environment, this may have influenced the recordings and potentially reduced differences between mood states. However, the effect of medications must be considered minimal, as there were minimal changes in prescribed medications between recordings”; after the sentence (line 500) “The current results of equal activity levels between mood states should be revisited taking these findings into account”.

2) The process that one-minute activity counts were derived from tri-axial acceleration data were not described. For example, some filtering was applied to the raw data?, how the 3-dimentional data were converted into one-dimensional data? The definition of “activity” is the most important in those research.

Answer: We thank you for identifying our missing description of the process for calculating the one-minute activity counts defining activity in the paper. The following description is added to the manuscript (line 144– 147 in the revised manuscript w/track changes): “The absolute mean of the 3-axis’ activity counts per minute was calculated for each time series of motor activity, by the formula | SQRT (x2 + y2 + z2) – Gravity|, then 1920 lines (the sum of 32 Hz multiplied by 60 seconds) were summed and divided by 1920. The calculated outputs are comparable to the motor activity data analyzed in previous studies of bipolar disorder (Scott et al., 2017, Tazawa et al., 2019, Krane-Gartiser et al., 2018, Fasmer et al., 2018 and Jakobsen et al., 2020).” 

3) The mathematical interpretation of the index, RMSSD/SD, might be provided for a potential reader of the journal. 

Answer: A proper mathematical interpretation of the RMSSD/SD index is now provided in the manuscript (line 180 -181 in the revised manuscript w/track changes). For a more comprehensive answer, see the answer to Comment no. 14 by Reviewer #2, as well as line 168-181 in the manuscript. 

4) Why was the string number set to six in the symbolic dynamics method?

Answer: The symbolic dynamics method is a method developed by Guzzetti (2005) and Porta (2001) as a pattern classifier supplement to complexity evaluations of heart rate variability time series. This method gives information about the uniqueness and distribution of patterns in a time series, while the complexity index Sample Entropy rates the degree of pattern repetitiveness in the time series. The symbolic dynamics method utilizes six strings, as increasing number of strings have been found to reduce stationarity in the time series. We have added a missing reference in the text; Porta A, Guzzetti S, Montano N, Furlan R, Pagani M, Malliani A, et al. Entropy, entropy rate, and pattern classification as tools to typify complexity in short heart period variability series. IEEE Transactions on Biomedical Engineering. 2001;48(11):1282-91.

5) Did the results change when a different number of cliques was selected?

 Answer: The number of k-cliques indicate how smooth the activity changes are. The more k-cliques the graph has, the smoother the activity changes are. In this paper, we have focused on 3-cliques. Most versions of the clique problem are hard. But the 3-cliques are computed with a O(a(G)m) time algorithm designed by Chiba and Nishizeki, where a(G) is the arboricity of the graph (Chiba N, Nishizeki T. Arboricity and Subgraph Listing Algorithms. SIAM Journal on Computing. 1985; 14(1):210–23.). By definition of similarity graph, every 3-clique indicates that a tuple of three nodes have similar values. By changing the number of cliques, we will find more information about data. However, as we explained above, it is a hard problem in Mathematics and Computer science to find all the cliques of a graph. Therefore, we focused on 3-cliques. We have also added the following phrase in the text: (We report on 3-cliques), “calculated by a method developed by Chiba & Nishizeki (1985)”.

6) Age often affects the activity levels or variance. The author may test the correlation of age with significant behavioral indices.

Answer: We recognize that age often affects the activity levels and variance. This topic is also discussed in the manuscript (line 540 – 542 in the revised manuscript w/track changes). However, as mentioned in the manuscript, the effect of age can likely be considered minor in the study, due to the within-subject design. On the other hand, testing correlation of age with significant behavioral indices would be both important and relevant when conducting a group comparison with motor activity data. 

7) Related above, the reviewer suggests to check the correlation among behavioral indices (e.g. sample entropy vs network properties). This may provide further insight on what kind of behavioral dynamics contributes the results obtained.

Answer: We greatly appreciated this suggestion from the reviewer, and have therefore expanded the results section of the paper with a Pearson correlation analyses (see line 383 – 392). We have also included a new table 5, presenting correlations between the estimates of variance, autocorrelation, complexity and the various outputs of the k = 2 similarity graph analysis, for manic evening results. This answer also relates to Comment no. 15 from Reviewer #2.

8) The results derived from 1190 minutes data were possibly affected by the differences in sleep between mania and euthymia. Sleep effects should be carefully treated.

Answer: We agree that sleep somehow probably has affected the difference between mania and euthymia in the 1190 minutes analyzes. However, in the analysis we have focused on the overall differences in the motor activity recordings. There has been no particular attention on sleep, beyond as a factor in the overall analysis. This approach is similar to other comparable studies, as well as systematic reviews on motor activity cited in the manuscript (Krane-Gartiser 2014, Scott 2017, Fasmer 2018, Krane-Gartiser 2018, Tazawa 2019, Fasmer 2020 & Jakobsen 2020). 

9) As the authors discussed by themselves, the results of the similarity graph method were inconsistent among different hyper-parameter values, or difference in time of day. This would be one of targets for criticism of this manuscript. Showing the systematic changes of behavioral indices with varying parameter vales may be one possible way to show the robustness of the results. More specifically, for example, calculate 3-cliques values for k=2,3,4,5,…, and then plot the results as a function of k. Similarly, effects of the choice the time window of data could be evaluated (e.g., 7:00-9:00, 8:00-10:00,….)

Answer: Differences in the results of similarity graph with various hyper-parameter values are to be expected, and in our opinion, do not question the robustness of the results. In fact, such differences are hypothesized to uncover subtle aspects in the activity of the underlying time series. To clarify our viewpoint, we have put in the follow sentence in the discussion (line 430 - 431 in the revised manuscript w/track changes):“On the other hand, these deviating results may be a revelation of subtle fluctuations in the morning activity patterns, undetectable when increasing the value of k (Fasmer et al., 2020).”

Regarding the time windows selected for morning and evening analyzes, we wanted the morning and evening time epochs to be 12 hours apart and minimally biased by circadian sleep-wake cycles. However (as written in the manuscript), “…. to make it possible to include data from most participants, with minimal missing data in the time series.” (Line 157 – 158 in the revised manuscript w/track changes). Consequently, the proposed evaluation of potential time windows becomes difficult due to missing data issues. However, we completely agree with the reviewer in that changing time window of data may adjust the outcomes, since being potentially influenced by both circadian/ultradian rest-activity cycles and social rhythms. 

To sum up, by changing parameters, we get different graphs, and different results.

Differences in time of day are also to be expected, due to changing circadian/ultradian rest-activity cycles and social rhythms. E.g., our method is not inconsistent.

10) Figure 2: no label and no unit on the y-axis

Answer: This has now been fixed. 

Reviewer #2: 

This is an interesting study that will be of interest to the readership of PLOS One and to the field of psychiatry more broadly. The authors identify within subject differences in the complexity of motor activity patterns between mania and euthymic states among individuals with bipolar disorder. Although the sample size is small, realistically it is difficult to collect such data and the authors should be commended for their effort. The study provides empirical support of what is clinically intuitive, using a sound methodology and the approaches taken are somewhat novel in the area. Some of the expression of terms in the manuscript is in my opinion quite awkward and could be improved. I have the following feedback that I believe the authors could integrate to improve the quality of this manuscript.

Abstract

1) - Line 37-38: the authors state that there were fewer edges and bridges in their analysis. It would be helpful for the reader if the authors provided some context in the abstract for people unfamiliar with this approach. As currently written, it assumes knowledge of this analysis before reading the rest of manuscript. Are edge and bridge frequency measures of complexity analogous to entropy? If so please state, if not please differentiate.

Answer: We agree with the reviewer, in that there are many graph definitions in abstract which may seem new to the readers. Therefore, we have inserted the following paragraph in the abstract (line 34 -39): “The similarity graph measures fluctuations in activity reasonably comparable to both variability and complexity measures. However, direct comparisons are difficult as most graph measures reveals variability in constricted time windows. Compared to sample entropy, the similarity graph is less sensitive to outliers. The little-understood estimate Bridges is possibly revealing underlying dynamics in the time series.”

Intro

2) - Line 54/55: needs citation.

Answer: A citation has been inserted; (6) Takaesu Y, Inoue Y, Ono K, Murakoshi A, Futenma K, Komada Y, et al. Circadian rhythm sleep-wake disorders as predictors for bipolar disorder in patients with remitted mood disorders. Journal of Affective Disorders. 2017;220:57-61. 

3) - Line 56: in my opinion ‘full-blown’ mood episode is a rather colloquial/unrefined way of describing a severe, clinically confirmed mood episode. The authors might consider revising.

Answer: Thank you for showing this rather colloquial wording; it has now been changed to “of a severity requiring hospitalization” 

4) - Line 71: ‘dissolved’ is a strange way to describe circadian dysfunction. I would suggest an attenuated or less robust circadian rhythm instead if that is what the authors mean? 

Answer: Thank you for showing this questionable wording and the description has been changed to “attenuated”, and similar wording has been changed in line 518 in the revised manuscript w/track changes. 

5) - Line 95/96: what do the authors mean when they state that non-linear dynamic analyses are considered the most rewarding method – could the authors revise the term ‘rewarding’. Do you mean recommended method or most informative method? You could additionally elaborate on why this method is superior.

Answer: We have replaced the word ‘rewarding’ with ‘useful’, besides elaborated why non-linear methos are superior. The following text has been inserted (line 98 – 101): 

“”Nonetheless, non-linear dynamic analyses are considered the most useful method to sufficiently disclose the information contained in motor activity (Scott J, Vaaler AE, Fasmer OB, Morken G, Krane-Gartiser K. A pilot study to determine whether combinations of objectively measured activity parameters can be used to differentiate between mixed states, mania, and bipolar depression. International Journal of Bipolar Disorders. 2017;5(1):5.). Simple linear models are found to be incapable of revealing the variability and complexity characterizing the activity patterns of bipolar disorder (Scott et al., 2017, Krane-Gartiser et al., 2014, Krane-Gartiser et al., 2018).” 

6) - General comments on introduction and positioning the study: the authors refer to the previous similar work conducted in this area (e.g. review of the Krane-Gartiser et al studies). However, it is not quite clear what the unique contribution of this study is. The authors suggest that previous studies have been conducted using small sample sizes, but so too is the present study one with a small sample size. The authors state that previous work has focused on groupwise differences and in contrast the current study involves a within-subject/intra-individual design. I believe the authors have missed the opportunity to emphasise how their approach advances the evidence demonstrated previously. Perhaps you might consider highlighting the advantages of intra-individual comparisons between acute and remitted mood states in order to communicate the novelty of your study.

Answer: We have followed the reviewer’s advice and inserted the following sentence in the manuscript (Line 82 - 83): “Especially for studies of change in motor activity related to change in mood state, a within subject design, where subjects are their own controls, are in demand (Jakobsen et al. 2020).” 

Method and Results

7) - The standard convention in the participants section would be to state the n of sample. This is done elsewhere in manuscript but should also be in the top-line of the participants section. 

Answer: This has now been stated, in line 112. 

8) - The brand and manufacturer of the actigraph used should be stated (or otherwise the manufacturer of the accelerometer that the researchers mounted to the wristband). Is reference 28 supposed to refer to this? Please make more explicit in methods section.

Answer: The reference refers to the brand and manufacturer of the actigraph. We used the Empatica E4 multisensor wristband, which is including an accelerometer. This information has now been entered into the text, at line 136. 

9) - Do I understand correctly that the accelerometry was only conducted over the space of one day? This is a major limitation and needs to be acknowledged. 

Answer: Thanks for asking this question. Recording motor activity for only 24-h was done due to battery life limitations of the Empatica E4 wristband. Still, we disagree in the reviewer’s opinion that this is a major limitation of the study. As all of the referenced review papers on actigraphic features in bipolar disorders have without any problems included studies analyzing 24-h motor activity recordings (De Crescenzo 2017, Scott 2017 and Tazawa 2019), we do not consider this a major limitation needing to be acknowledged.

10) - Were there any bespoke R packages used to analyse these data? If so it would be best practice to acknowledge and reference. Also is a markdown of this R code available anywhere for those who wish to apply a similar approach?

Answer: No R packages was used to analyze our data. R was only applied to process the raw 32 Hz motor activity data. For further details, see the answer to Reviewer#1 comment no. 2. 

11) - The authors should provide further justification for selected the morning and evening time windows. Was this decided post-hoc upon inspection of available data (implied in manuscript) or was this an a priori decision based on some inference of the circadian rest-activity pattern? If the latter the authors should be aware that the biological morning and evening is determined by the underlying circadian rhythm and, as this was not assessed, arbitrarily defining morning and evening times for the purpose of comparing circadian timing is not appropriate. Later in the discussion the authors use these findings to argue that there is a reduction in circadian variation which I do not think is appropriate. They looked at a diurnal difference within a two hour period in both the morning and evening… any claims made about the underlying circadian rhythm of patients with mania should be revised and the conclusions tempered.

Answer: We agree with the reviewer in that further justification for selecting the morning and evening time series need to be provided. Therefore, the current text in the method chapter has been extended to the following: “The time series were decided post-hoc, upon inspection of available data. The criteria for selection of the specific time series was; being 12 hours apart and minimally biased by circadian sleep-wake cycles. Then, to make it possible to include data from most participants, with minimal missing data in the time series.”

 (Line 154 – 158 in the revised manuscript w/track changes). 

We totally agree with the reviewers’ statement about circadian timing. Nevertheless, as this study aims to characterize activity patterns of the bipolar manic state by comparing manic patients intra-individually to their euthymic selves, we chose a similar approach to previous studies comparing manic patients to healthy controls (Krane-Gartiser et al., 2014). However, we do think that the results of comparing morning and evening within state and individual (table 4) both legitimate and requires a discussion related to sparse evidence and hypothesis regarding the circadian rhythmicity of the bipolar manic state. At the same time, we realizes that our wordings about the underlying circadian rhythm of patients with mania are somewhat grandiloquent, and we have therefore both revised and moderated our formulations in the discussion and the conclusion. (Line 506-507, 515, 535 and 611-614 in the revised manuscript w/track changes).

12) - Regarding the threshold of <5% missing data considered acceptable – do the authors have a reason for selecting this minimum threshold or a source they can reference for this recommendation? 

Answer: In the literature, it has been postulated that missing data of 5 % or less is inconsequential. We have now inserted a reference for the recommendation (Dong Y, Peng C-YJ. Principled missing data methods for researchers. Springerplus. 2013;2(1):222-.) in line 161. 

13) - Why was autocorrelation with a lag of 1 minute chosen? The Krane-Gartiser study to which the authors refer in discussion used several different autocorrelation lag windows. 

Answer: Autocorrelation at lag 1 is an estimate that has been applied in research in critical transitions preceding tipping points before shifts in dynamic system (Dakos, V., van Nes, E.H., D'Odorico, P. and Scheffer, M. (2012), Robustness of variance and autocorrelation as indicators of critical slowing down. Ecology, 93: 264-271.) 

In our analyses, the 1 lag delay equals one minute, due to features (the absolute mean of the 3-axis’ activity counts per minute) of the data. The Krane-Gartiser (2014) study had a similar approach, and did only use autocorrelation at lag 1 in their analyses. In this case, the reviewer must somehow have misunderstood the Krane-Gartiser papers approach. Anyway, we have extended the text: “..., and is a common method applied within dynamic system research (Dakos et al., 2012), in line 185.

14) - For the general reader who is unfamiliar with the RMSSD measure it might be useful to state that this measures instability between successive intervals rather than solely variability as assessed by SD. Furthermore, it is unclear to me why both are given as a ratio of the mean. Is there an assumption that greater motoric activity is correlated with both greater RMSSD and SD and the ratios are an attempt to deal with this confound? (e.g. deriving a coefficient of variation). Similarly, the reason for using the RMSSD as a ratio to SD could be explained. Could the authors please expand on their methods here. Is this to uncouple epoch-to-epoch instability from over variability?

Answer: We completely agree with the reviewers that the estimates of variance generally need a better explanation. To answer the reviewer’s questions and demands we have updated the first paragraph of the Mathematical Analyses subchapter line 168 – 181 in the revised manuscript w/track changes), and inserted the following text: “Standard deviation (SD) is a measure of how dispersed the data are in relation to the mean. Low standard deviation means data are clustered around the mean, and high standard deviation indicates data are more spread out. The standard deviation is calculated as the square root of variance by determining each data point's deviation relative to the mean. The coefficient of variation (CV) is obtained by dividing SD to the Mean. It describes the variability of a sample relative to its mean. This measure is unitless, expressed as a percentage, and recommended applied in time series with unstable means (Reed et al., 2002). In our experience, this definitely applies to time series of motor activity. Therefore, in this paper, SD is in fact CV. The root mean square successive differences (RMSSD) is the root mean square of successive differences between all the time epochs, and indicates how much a set of data varies within itself (Von Neumann et al., 1941). For the same reason as for SD, RMSSD is given as a ratio to the mean. Finally, the RMSSD/SD ratio was calculated. Because RMSSD provides the variability between successive intervals instead of solely variability, as assessed by SD, the RMSSD / SD ratio reveal how scattered the data is in itself.” 

15) - The authors go into great detail in explaining the similarity graph algorithm. However, I believe they should highlight what this measure reflects and how it different from variability, instability, and complexity as assessed by the other measures. What is the benefit of this analysis and if similar to the entropy and symbolic dynamics method? Could the authors provide context on why this does not create a redundancy of measures?

Answer: As a general answer to the reviewer’s questions, we have updated the text comparing the similarity graph to variability and complexity measures; see line 449 to 470 in the revised manuscript w/track changes. Furthermore, we agree with the reviewer’s assertion that our approach appears to create a redundancy of measures. However, as the dataset presented in this paper is rather unique and has not been previously studied, we have analyzed the data with the commonly applied linear and non-linear mathematical models to be expected. The similarity graph algorithm, as a rather new and experimental method within the field of studying the motor activity of bipolar disorder, has been added due to the optimistic discriminating abilities presented in two previous papers from our research group (Fasmer 2018 and Fasmer 2020). The major uniqueness of this study is the presentation of a within-subject dataset of manic/euthymic bipolar patients. The secondary uniqueness is related to the properties of the similarity graph algorithm. Furthermore, regarding the worries about redundancy of measures, we have expanded the results section of the paper with a Pearson correlation analyses. We have also included a new table 5, presenting correlations between the estimates of variance, autocorrelation, complexity and the various outputs of the k = 2 similarity graph analysis, for manic evening results. This answer also partly relates to Comment no. 7 from Reviewer #1.

16) - Table 2 correction: for the RMSSD (% of mean), the value should be 69.8 rather than using a comma as a decimal point. 

Answer: Thanks for spotting the error, which now has been corrected. 

17) - Several results in Table 2 and Table 3 are reported as significant at P<0.05 yet the methods section describes a Bonferroni revised alpha of 0.0125 being the threshold for statistical significance. Please clarify which is the correct threshold and update the results and discussion accordingly.

Answer: In the paper, we employ both a significance level of p < 0.05 and p < 0.0125, depending on the features of the conducted analyzes. To clarify the reasons and the rationale for this, we have updated the text in the manuscript slightly: “A p-value < 0.05 was considered statistically significant when comparing mood states in table 2 and 3. When comparing mania and euthymia within both state and subject in table 4, we adjusted the p-value according to a Bonferroni correction for multiple comparisons to avoid a type 1 error (Lee & Lee, 2018). For these analyses, a p-value less than 0.0125 was considered statistically significant.” 

Discussion

18) - Generally, the discussion could be improved by situating the novelty of the authors findings within the theme of how these findings might be clinically translated. Much of the discussion section in which the authors interpret their results recaps the results section.

Answer: We have added the following sentences at the beginning of the discussion:

“The novelty of this study is that our findings provide empirical support of how the agitated manic psycho motoric activity differs from euthymia intra-individually in humans with bipolar disorder. In the first place, the clinical relevance of this study may appear to be of minimal importance; however, as an evidence-based foundation for future innovations in clinical aids, the clinical relevance of this study may be of significant importance.”

19) - The use of circadian variation in lines 437-438 is not appropriate for the reasons I have previously detailed. Caution should be applied in making inferences about the circadian timing system as there is no contemporaneous measure of endogenous clock function. Please give consideration to this point in the revised manuscript.

Answer: This comment has previously been answered in Comment 11. We totally agree with the reviewer’s suggestions, and the sentence has been changed to: “When comparing morning and evening recordings within states, we potentially discovered what might appear to be reduced circadian variation in the manic state.” (Line 506-507 in the revised manuscript w/track changes)

20) - Anitpsychotic and mood stabiliser medications are mentioned in the discussion, but these are not highlighted as limitations. Both may have profound effects on motor rest-activity patterns and surely obfuscate the effect of mood state.

Answer: We agree on your statement that the antipsychotic and mood stabilizing medications influence the motor activity, and that this fact probably affects our data (as discussed in the paper). Still, we do not agree that these medications should be highlighted as a limitation, since the prescribed medications did not differ between mood states. On the other hand, if we had compared a group of manic patients to a group of healthy controls, similar to done in the Krane-Gartiser (2014) paper, the medications in the patient group definitely becomes a limitation. As an argument for our view, we have inserted the following sentence in the text (line 504 – 505): “However, the effect of medications must be considered minimal, as there were minimal changes in prescribed medications between recordings.”

21) - The authors state that no previous studies have applied such state-of-the-art measures. Yet, as I understand it, the novelty is limited to the similarity graph algorithm. Most of the measures of instability/complexity/autocorrelation have previous been applied in other studies. Therefore, the authors should revise this claim.

Answer: The major uniqueness of this study is the presentation of a within-subject dataset of manic/euthymic bipolar patients. The secondary uniqueness is related to the properties of the similarity graph algorithm. The dataset presented in this paper is rather unique and has not been previously studied. Therefore, we have analyzed the data with the commonly applied linear and non-linear mathematical models. The similarity graph algorithm, as a rather new and experimental method within the field of studying motor activity in bipolar disorder, has been added due to the optimistic discriminating abilities presented in two previous papers from our research group (Fasmer 2018 and Fasmer 2020).

---

## [Decision Letter · Decision Letter 1]

5 Nov 2021

PONE-D-21-23669R1Complexity and variability analyses of motor activity distinguish mood states in Bipolar DisorderPLOS ONE

Dear Dr. Jakobsen,

Thank you for submitting your manuscript to PLOS ONE. After careful consideration, your work should be revised further to fully meet PLOS ONE’s publication criteria. Therefore, we invite you to submit a revised version of the manuscript that addresses the points raised during the review process.

In particular, the role and impact of sleep pattern on your findings should be carefully evaluated and reported. We recommend to remove the sleep period data and re-evaluate the experimental results.

Once you do so, please submit your revised manuscript by Dec 20 2021 11:59PM. If you will need more time than this to complete your revisions, please reply to this message or contact the journal office at plosone@plos.org. Please include the following items when submitting your revised manuscript:A rebuttal letter that responds to each point raised by the academic editor and reviewer(s). You should upload this letter as a separate file labeled 'Response to Reviewers'.A marked-up copy of your manuscript that highlights changes made to the original version. You should upload this as a separate file labeled 'Revised Manuscript with Track Changes'.An unmarked version of your revised paper without tracked changes. You should upload this as a separate file labeled 'Manuscript'.If applicable, we recommend that you deposit your laboratory protocols in protocols.io to enhance the reproducibility of your results. Protocols.io assigns your protocol its own identifier (DOI) so that it can be cited independently in the future. For instructions see: https://journals.plos.org/plosone/s/submission-guidelines#loc-laboratory-protocols. Additionally, PLOS ONE offers an option for publishing peer-reviewed Lab Protocol articles, which describe protocols hosted on protocols.io. Read more information on sharing protocols at https://plos.org/protocols?utm_medium=editorial-email&utm_source=authorletters&utm_campaign=protocols.

We look forward to receiving your revised manuscript.

Kind regards,

Gaetano Valenza, Ph.D.

Academic Editor

PLOS ONE

Journal Requirements:

Reviewers' comments:

Reviewer's Responses to Questions

**Comments to the Author**

1. If the authors have adequately addressed your comments raised in a previous round of review and you feel that this manuscript is now acceptable for publication, you may indicate that here to bypass the “Comments to the Author” section, enter your conflict of interest statement in the “Confidential to Editor” section, and submit your "Accept" recommendation.

Reviewer #1: (No Response)

Reviewer #2: All comments have been addressed

2. Is the manuscript technically sound, and do the data support the conclusions?

Reviewer #1: Partly

Reviewer #2: Yes

3. Has the statistical analysis been performed appropriately and rigorously? 

Reviewer #1: Yes

Reviewer #2: Yes

4. Have the authors made all data underlying the findings in their manuscript fully available?

Reviewer #1: Yes

Reviewer #2: Yes

5. Is the manuscript presented in an intelligible fashion and written in standard English?

Reviewer #1: Yes

Reviewer #2: Yes

6. Review Comments to the Author

Reviewer #1: Thank you for your responses to my comments and suggestions. But, I still have concern about the effects of sleep on your results. To my comment # 8, the authors respond `in the analysis we have focused on the overall differences in the motor activity recordings. There has been no particular attention on sleep, beyond as a factor in the overall analysis’.

As clearly shown in Fig. 2, activity pattens and activity mean levels during sleep periods are different between mania and euthymia. I suspect that those clear differences have non-negligible effects on the results. The authors may discuss this or examine effects of activity data during sleep on their results, for example, by removing sleep period data.

Reviewer #2: The authors have returned a considered response to each of the points raised and this revision improves the overall quality of the original manuscript. I am pleased to recommend publication.

7. PLOS authors have the option to publish the peer review history of their article (what does this mean?). If published, this will include your full peer review and any attached files.

Reviewer #1: No

Reviewer #2: No

---

## [Author Response · Author response to Decision Letter 1]

17 Dec 2021

Thank you for asking this question regarding your concerns about the effects of sleep on our results. After rethinking the matter, we fully agree that it is important to consider if sleep periods may have substantially affected our results. Still, regarding the 1190 minutes analyzes our response is similar to what we responded to your comment # 8 last time; “In the 1190 minutes analysis we focus on the overall differences in the motor activity recordings, therefore there will be no particular attention on sleep, beyond as a factor in the overall analysis.” It must be pointed out that our viewpoint is in line with how the motor activity of bipolar patients are proposed analyzed by a systematic review (Scott et al. (2017) Activation in bipolar disorders: A systematic review. JAMA Psychiatry), as well as in a comparable study of mania (Krane-Gartiser et al. (2014) Actigraphic assessment of motor activity in acutely admitted inpatients with bipolar disorder. PloS one.) 

However, regarding the two shorter 120-minute periods, we have this time removed all sleep period data from the analyses, by adjusting the time series according to the current sleep/wake pattern of each individual time series. We followed a comparable methodology for the selection of morning and evening periods to Krane-Gartiser and co-workers (et al. 2014) method for analyzing similar patients. By such an approach, we have also been able to include data from two more participants of the clinical study. The procedure for our new approach is presented in the subchapter “Recordings of Motor Activity” in the “Materials and methods” part. The results changed, as we this time found more of the variables significantly different in the morning (similarity graph estimates, sample entropy and symbol dynamics), and less variables significantly different in the evening, when comparing mania to euthymia. However, when comparing the morning and evening periods within mood states we found no significant differences for neither mania nor euthymia. The changed results led to some modifications presented in the Results, Discussion and Conclusion parts of the paper. Finally, we also updated the Pearson correlation analyses for the manic variables in accordance with the new results.

---

## [Editor Report · Decision Letter 2]

20 Dec 2021

Complexity and variability analyses of motor activity distinguish mood states in Bipolar Disorder

PONE-D-21-23669R2

Dear Dr. Jakobsen,

We’re pleased to inform you that your manuscript has been judged scientifically suitable for publication and will be formally accepted for publication once it meets all outstanding technical requirements.

Kind regards,

Gaetano Valenza

Academic Editor

PLOS ONE

---

## [Editor Report · Acceptance letter]

23 Dec 2021

PONE-D-21-23669R2 

Complexity and variability analyses of motor activity distinguish mood states in Bipolar Disorder 

Dear Dr. Jakobsen:

I'm pleased to inform you that your manuscript has been deemed suitable for publication in PLOS ONE. Congratulations! Your manuscript is now with our production department. 

Kind regards, 

on behalf of

Dr. Gaetano Valenza 

Academic Editor

PLOS ONE